# Identifying Clinicoradiological Phenotypes in Diffuse Idiopathic Skeletal Hyperostosis: A Cross-Sectional Study

**DOI:** 10.3390/medicina57101005

**Published:** 2021-09-24

**Authors:** Teresa Clavaguera, Patrícia Reyner, Maria Buxó, Marta Valls, Eulàlia Armengol, Xavier Juanola

**Affiliations:** 1Servei de Reumatologia, Hospitals Universitari Josep Trueta i Hospital, Santa Caterina, 17007 Salt, Spain; patricia.reyner@ias.cat (P.R.); marta.vallsr@gencat.cat (M.V.); 2Servei de Biostadística, Institut d’Investigació Biomèdica de Girona (IDIBGI), 17003 Girona, Spain; mbuxo@idibgi.org; 3Hospital de Palamós, Palamós, 17230 Girona, Spain; euarmengol@gmail.com; 4Hospital Universitari de Bellvitge, Hospitalet de Llobregat, 08907 Barcelona, Spain; x.juanola@bellvitgehospital.cat

**Keywords:** diffuse idiopathic skeletal hyperostosis, enthesopathy, ossification, heterotopic, phenotype, ankylosing hyperostoses, vertebral, ankylosing spondylitis

## Abstract

*Background and objectives*: Diffuse idiopathic skeletal hyperostosis (DISH) is a bone formation disease in which only skeletal signs are considered in classification criteria. The aim of the study was to describe different phenotypes in DISH patients based on clinicoradiological features. *Materials and Methods*: We evaluated 97 patients who met the Resnick or modified Utsinger classification criteria for DISH and were diagnosed at our hospital from 2004 to 2015. Patients were stratified into: (a) peripheral pattern (PP)—Resnick criteria not met but presenting ≥3 peripheral enthesopathies; (b) axial pattern (AP)—Resnick criteria met but <3 enthesopathies; and (c) mixed pattern (MP)—Resnick criteria met with ≥3 enthesopathies. Statistical analysis was carried out to identify variables that might predict classification in a given group. *Results***:** Fifty-six of the 97 patients included (57.7%) were male and 72.2% fulfilled the Resnick criteria. Applying our classification, 39.7% were stratified as MP, 30.9% as AP and 29.4% as PP. Clinical enthesopathy was reported in 40.2% of patients during the course of the disease. Sixty-eight patients were included in a comparative analysis of variables between DISH patterns. The results showed a predominance of women (*p* < 0.004), early onset (*p* < 0.03), hip involvement (*p* < 0.003) and enthesitis (*p* < 0.001) as hallmarks of PP. Asymptomatic patients were most frequently observed in AP (28.6%, MP 3.8%, PP 5.0%) while MP was characterized by a more extensive disease. *Conclusions*: We believe DISH has distinct phenotypes and describe a PP phenotype that is not usually considered. Extravertebral manifestations should be included in the new classification criteria in order to cover the entire spectrum of the disease.

## 1. Introduction

Diffuse idiopathic skeletal hyperostosis (DISH), or Forestier-Rotés disease, is characterised by the calcification and ossification of the entheses, the anchoring area for fasciae, ligaments, tendons and joint capsules in the bone [1]. The hallmark of the disease is ossification of the anterior common vertebral ligament of the dorsal column, a process that can extend to the rest of the vertebral sites, as well as to multiple peripheral entheses [2,3]. In spite of this, the Resnick classification criteria (which remain the most widely used) only take account of axial manifestations [4].

In daily practice, there is a significant group of patients who do not present Resnick criteria but suffer a hyperostotic process that is not attributable to other rheumatic conditions. Some present fewer than three bridges of ossification; others have significant enthesophytes that do not form consecutive bridges, or few vertebral signs but prominent peripheral enthesophytes; in other cases, the presence of characteristic enthesophytes in the acetabular side of the hip joint are the main feature, rather than axial involvement. In a previous study, we reported three patients with an extensive ossification of entheses who did not fulfil the Resnick criteria, but who were indisputably affected by hyperostosis. We noted a distinct phenotype with predominant extraspinal manifestations, female predominance and earlier disease onset [5]. Recently, some authors have suggested that peripheral manifestations, such as hip involvement, could be prominent in DISH and lead to significant discapacity or functional impairment [6]. Hawang et al. reported a set of 421 hip arthroscopies, in which 17 patients had characteristic hip involvement in DISH, but only nine had significant spinal changes [7].

For this reason, other authors have proposed including the involvement of peripheral entheses in a new set of classification criteria that covers the disease spectrum entirely [8,9]. Utsinger described it as a symmetric enthesopathy predominantly in the calcaneus, olecranon or patella with the formation of new bone with a well-defined cortical margin [10].

However, no consensus emerged with regard to either the inclusion or exclusion of extraspinal features or the number of bridges necessary to establish an accurate DISH classification [11,12]. A panel of experts reviewing the standard criteria, reached a consensus on spinal features, but not on the inclusion of peripheral enthesopathies or metabolic disorders [13]. Thus, these criteria allow for diagnosis at an advanced state of the disease. In this situation, a way must be found to diagnose the disease at early stages and to define “early-DISH”, as some authors are attempting to do at present [14]. To determine whether extraspinal features should be included, it must first be established whether there are patients with mild or no spinal involvement but characteristic peripheral symptoms. If so, and if these patients are excluded from the DISH evaluation, specific morbidity will be overlooked.

Based on our preliminary observations, we hypothesised that different patterns or phenotypes can be distinguished in DISH subjects; we defined three: peripheral pattern (PP), axial pattern (AP) and mixed pattern (MP), based on the predominance of vertebral or extravertebral signs (peripheral entheses). Our objective was to assess these patterns in our patient cohort and to analyse distinct aspects of each cluster.

## 2. Methods

We conducted a cross-sectional study of our historical cohort of DISH patients to determine the presence of the phenotypes that we proposed above. This observational study was performed in accordance with the recommendations of the ‘Strengthening the Reporting of Observational studies in Epidemiology’ (STROBE) guidelines [15].

Our cohort was recruited from a database administered by Palamós Hospital (Girona, Spain), which serves a suburban area of 132,906 inhabitants, between January 2004 and September 2015. This database provides encrypted patient identification numbers, gender information, date of birth and the International Classification of Diseases, Ninth Revision, Clinical Modification (ICD-9-CM) codes for diagnoses. We selected subjects codified as DISH (ICD-9-CM code 721.6). Although the diagnosis was established by specialists in rheumatology, orthopaedics and rehabilitation at the Musculoskeletal Unit, an experienced rheumatologist (T.C.) checked that all criteria were properly met. She also examined all X-rays to select patients whose medical records contained the necessary radiological scans for statistical analysis. No X-rays were requested solely for the purposes of this study.

We recorded patients’ data from the visit when their diagnosis was established. A radiological study was requested of the X-rays carried out for the four years preceding the diagnosis. Diagnosis was confirmed if patients met both Resnick’s and Utsinger’s criteria. Resnick’s classification is based on the presence of flowing calcification and ossification along the anterolateral aspects of at least four contiguous vertebral bodies, with relative preservation of intervertebral disc height of the vertebral segments involved. Additional requirements were the absence of extensive radiographic changes associated with degenerative disc disease, as well as the absence of apophyseal joint bony ankylosis and sacroiliac joint erosions, sclerosis or bony fusion. It is important to stress that Utsinger lowered the threshold for spinal involvement to two contiguous vertebral bodies and added the presence of peripheral enthesopathies to the diagnostic criteria.

In the general cohort, we included patients whose medical records contained at least one X-ray of a single vertebral location, in tandem with a sacroiliac image, which are sufficient not only to fulfil the Resnick requirements but also to establish a differential diagnosis of axial spondyloarthritis (AS). However, for analysing the clinicoradiological variables with regard to the patterns, we also required the complete column images (anteroposterior and lateral), the pelvis (anteroposterior) and at least three entheseal sites (trochanter, knees, elbows and feet). One experienced rheumatologist (T.C.) reviewed all X-rays.

Furthermore, patients were excluded from the study if they had: (a) personal or first-degree history of axial or peripheral spondyloarthritis, (b) positive HLAB27, (c) personal or first-degree history of psoriasis or inflammatory bowel disease or uveitis, even if they otherwise fulfilled the diagnostic DISH criteria. If any of these conditions was not recorded in the medical record, the subject was excluded.

Based on previous observations, we proposed three clinical-radiological patterns: (a) peripheral pattern (PP) for patients who did not fulfil the Resnick criteria but who presented ≥3 peripheral enthesopathies (probable or possible Utsinger categories) with or without vertebral spine manifestations; (b) axial pattern (AP) for patients who met the Resnick criteria but who presented <3 peripheral enthesopathies; and (c) mixed pattern (MP) for patients who met the Resnick criteria but who presented ≥3 peripheral enthesopathies (Figure 1).

Demographic data such as age and gender were collected from the patient’s medical records, as were the following baseline comorbidities: arterial hypertension (ICD-9-CM codes 401,405), cardiovascular events (coronary heart disease (ICD-9-CM codes 410–414 and/or cerebrovascular disease 430–438) and dyslipidaemia (ICD-9-CM code 272). These comorbidities were included because they had previously been studied in DISH populations.

We defined the presenting symptom as the first complaint related to DISH, even though it had not led to diagnosis at that time. Hip involvement was recorded when we detected a significant ossification of the superior and/or inferior acetabulum margins with no joint space narrowing [16]. Our definition of peripheral enthesopathy referred to an ossification in a particular entheseal area, but also included exuberant unilateral ossifications as indicated by an Aydin’s semiquantitative score of 2 or 3 (moderate or severe) [17]. Otherwise, we defined clinical enthesopathy as a symptom when it was clinically relevant (pain with or without swelling at one of the entheses) and was codified in the medical record (ICD-9-CM codes 726) in a place with a prominent enthesophyte. We analysed the following locations: hips (greater trochanter), knees (superior and inferior patella, tibial tuberosity), ankles (Achilles tendon insertion), feet (the plantar aspect of calcaneus) and elbows (olecranon) according to the specific anatomic locations described in the literature [18]. We graded enthesopathy as present or absent. With respect to signs in the spine, we recorded the absence (unaffected) or presence (affected) of ossifications with or without incomplete or complete bridging of the disc space in any vertebral level.

The study was approved by the local ethical and clinical research committees. (HEDIPAT Protocol number V1: 07/11/2019).

## 3. Statistical Analysis

Variables are presented as the mean (standard deviation, SD) for continuous data and as frequencies and percentages for categorical data. The Kolmogorov–Smirnov test assessed the normality of continuous data distribution before statistical analysis. Differences between clinical-radiological patterns (PP, AP and MP) and categorical variables were analysed using chi-square or Fisher’s exact test, as appropriate. For continuous variables, comparisons between patterns were performed using one-way analysis of variance (ANOVA) with post-hoc Bonferroni’s test. All tests were 2-sided, and a level of *p* < 0.05 was considered statistically significant. Missing values were not imputed. Data management and all statistical analyses were performed using IBM SPSS 25.0 statistics software (IBM, Armonk, NY, USA).

## 4. Results

We recruited 167 patients affected by DISH, but only 97 were eligible for the purpose of our study. The flow chart of patient enrolment is shown in Figure 2, and Table 1 displays the characteristics of the study sample. The participants were predominantly male (57.7%) with a mean age (S.D.) at diagnosis of 65.6 (9.5) years. The mean age at onset was 58.2 (9.0) years, although 61.9% were under 65 years old; in fact, 16.5% experienced their first symptom before age 50. Regarding comorbidities, 51% had hypertension, 25% had Type 2 diabetes and 11.5% had suffered from a cardiovascular event. In all, 27.8% of patients fulfilled the Utsinger, but not the Resnick, criteria. However, 7.2% presented minimal or no axial involvement (DISH Criteria 3 by Utsinger).

Analysing the cohort separately according to gender, 69.4% of men met the established criteria for DISH, but only 36% of women did. We recorded a variety of initial symptoms of the disease, with pain or limited thoracolumbar spine mobility being the most frequent (43.8%). Clinical enthesopathy (24%), hip involvement (5.2%) or cervical complaints (16.7%) were other initial manifestations, while 10.4% of patients were asymptomatic at the time of diagnosis (these latter patients were classified based on radiographic scans requested for other reasons). Although clinical enthesopathy (40.2%) was frequently present during the course of the disease, it was not always recorded as a DISH-related symptom in the medical records. In our cohort, the distribution was divided as follows: PP 29.4%, AP 30.9% and MP 39.7%.

With regard to radiological signs, axial involvement was located in 92% of the thoracic spine, 57.8% of the cervical spine and 56.2% of the lumbar spine. Hyperostosis of the acetabular rim was found in more than half of patients. The entheses most frequently recorded were the trochanteric sites (81.3%) followed by Achilles insertion (79.6%) and plantaris fascia (75%); however, these data must be treated with caution as not all subjects underwent complete radiologic examinations.

We then conducted a bivariate analysis to determine whether there were any relevant differences between clinical and radiological variables with regard to specific patterns (Table 2). Sixty-eight subjects were eligible for the analysis, since no additional scans were requested. All patients included had a sufficient radiographic study as described in the Methods section.

There was a notable predominance of female patients in the PP group compared to the AP group *(p =* 0.004). Although the age distribution was homogeneous, by analysing ages at disease onset, we found that PP patients were diagnosed at a younger age than the rest (*p* = 0.036). Rates of delayed diagnosis were similar across all three phenotypes, though the delay tended to be longer in the PP group. We also investigated comorbidities such as diabetes, hypertension, dyslipidemia and cardiovascular events, but only observed an increased number of cardiovascular events in the AP group (*p* = 0.001).

The initial symptom recorded also showed statistically significant differences, although the most pronounced was clinical enthesopathy, which established a distinction between PP and AP subjects (*p* = 0.007). Another interesting finding was that 28.6% of AP cases were diagnosed based on radiological signs in an otherwise asymptomatic subject, versus 5% and 3.8% in the other two phenotypes; however, these results did not reach statistical significance. Hip complaints in PP (10%), and cervical spine involvement in MP (19.2%), also presented slight differences with regard to other groups at disease onset. For its part, clinical enthesopathy throughout the course of the disease was the main manifestation in PP (85%), though it was also reported in the other two patterns (AP 23.8%; MP 40.7%) (*p =* 0.001).

According to radiolographical data, the thoracic spine was the most commonly affected location, although 21.1% of PP patients had no ongoing evidence of hyperostosis at that level. Cervical and lumbar enthesophytes were also more frequent in MP and AP (*p =* 0.001). Hip excrescences in the acetabular rim were more common in PP (65%) and MP (58.3%) compared to AP, in which only 26.3% showed this radiological sign (*p =* 0.035).

## 5. Discussion

This is the first study to assess different phenotypes in DISH. We hypothesised that the disease would show different patterns based on both clinical symptoms and radiologic findings, especially including extraskeletal manifestations. Based on our proposal, we identified and described three different patient profiles.

The first is an AP pattern that showed a preponderance of spinal manifestations with limited extravertebral signs. These cases displayed male predominance, older age at the time of diagnosis and higher rates of cardiovascular events. Symptoms related to the thoracolumbar spine mainly occurred at disease onset, though it was also possible to establish a diagnosis of DISH based on radiological images in asymptomatic patients. This group met all the Resnick criteria, but as their radiological signs were sometimes limited to the thoracolumbar spine, we considered this to be the least symptomatic pattern.

The second is MP, characterised by an extensive osteoformation process that encompasses a large number of vertebral bodies and extraspinal entheses. Here, we also noted a male predominance, albeit to a smaller degree, and a mean age at diagnosis similar to AP. These patients complained of spinal or extraspinal symptoms at the same proportion, and disease diagnosis on the basis of radiological findings alone was unusual. Cervical spine ossification was more frequently reported in this phenotype and may represent a hallmark condition.

Finally, PP patients were distinguished by a higher preponderance of women, younger ages at the time of diagnosis, and extensive extraspinal involvement with limited or absent vertebral manifestations. As a hallmark of this cluster, clinical enthesopathy was the most frequently recorded initial symptom. In addition, hip involvement was more often reported, especially compared to the AP pattern. By definition, these patients do not fulfil the Resnick classification criteria. We are conscious that the presence of enthesophytes can be physiological and are especially related to patients’ age and their activity or work; Guldberg-Moller et al. [19] recently reported more enthesophytes in distal quadriceps and Achilles insertions that might be related to traction. These observations provide interesting data for future investigations in order to determine which enthesopathies are most significant or representative of DISH. Consequently, we considered that three or more enthesis sites were necessary to classify a PP pattern bearing in mind the exclusion criteria. We also found that 65% of patients with this phenotype presented a characteristic hip involvement that should be taken into account in order to assess the possibility of an “extraspinal DISH” with scarce axial involvement.

In the light of our data, we feel that there are certain aspects of the disease that should be reconsidered. First, the exclusion of extravertebral symptoms loses an essential part of the disease’s clinical spectrum and accurate evaluation of functional status impairment. Initially several authors debated whether or not DISH represented a true disease per se, or merely a morphological state characterised by excessive osteoformation [4,20,21,22]. Mata and colleagues published an important study on clinical manifestations and functional status in DISH patients, comparing them with lumbar spondylotic patients and healthy subjects [23]. They observed that lumbar pain and stiffness was more common in spondylosis, but conversely, they reported similar rates of cervical, thoracic and lower limb pain. Moreover, they noticed a higher frequency of upper limb symptoms related to enthesopathy and dysphagia in hyperostotic patients. The authors concluded that DISH was also a significant cause of disability, but the broad spectrum of the disease needs to be evaluated.

In relation to Mata’s observations, we noted that clinical enthesopathy is not solely limited to PP, since more than 20% of AP and 40% of MP subjects presented this symptom during the course of the disease. We consider that a patient with pain and significant bony excrescences should be diagnosed as having DISH, even if they lack many spine ossifications. For the same reason, we recommend that cervical spine and hip scans be included in the diagnostic process. DISH has a wide range of clinical symptoms that need to be considered in their totality.

On the other hand, some patients with DISH are asymptomatic at the time of diagnosis: 10% in our general cohort, but up to 28% of AP patients. According to this observation and Mata’s findings, we consider that DISH studies using only thoracolumbar X-rays or CT as part of their enrolment criteria could create a selection bias of a group of less symptomatic patients.

Another point of discordance was gender distribution. Classically, published data based on the Resnick classification criteria reported a high male predominance, about five to seven times higher than in women [24,25]. In our cohort, we found a more homogeneous gender distribution (1:1), but when we analysed prevalences, in respect to a specific pattern, the results were entirely different; males predominated in AP, but females in PP. These results raise the question of whether women are frequently underdiagnosed in DISH. Furthermore, we also noticed some gender-related radiological differences: males usually met the Resnick criteria while most women did not. Indirectly, we assumed that men had more axial structural damage, although we did not use any radiographic-specific tools to compare the cohorts. Several studies in AS have reported gender-based radiographic differences, in so far as women tended to have less structural damage [26,27]. We cannot reach any conclusions about extravertebral ossification gender differences due to the lack of complete extravertebral X-ray studies. Returning to AS and focusing on extravertebral involvement, Aydin et al. analysed the relationship between axial osteoformation, as measured by the modified Stoke Ankylosing Spondylitis Spine Score (mSASSS), and enthesophyte formation in Achilles tendons as measured by ultrasound [17]. The authors concluded that there was a significant correlation between the degrees of spinal and peripheral osteoformation. They also observed a strong association with male gender and proposed a bone forming gender-specific phenotype, which would be interesting to assess in DISH.

Furthermore, DISH has been associated with many metabolic disorders as part of the spectrum of the metabolic syndrome that can lead to major cardiovascular disorders [28]. Despite the characteristics of our studied population, the prevalence rates of diabetes, as well as those of hypertension, dyslipidaemia and cardiovascular events, are similar to those reported in the literature [29,30,31]. We did not find any differences between phenotypes, with the exception of cardiovascular events in the AP group; however, the latter result should be considered with caution because the small sample size could lead to bias.

The present study has some limitations. It is a descriptive study of our historical cohort, with the restrictions that this entails. Our aim was to establish a starting point for future investigations that include all features of the DISH spectrum. Moreover, not all physicians who referred the patients to us considered DISH when extraspinal features predominated without significant spine signs; therefore, the pattern prevalences are not reliable. We also stress that the Utsinger criteria require symmetrical enthesopathies, but we included some patients with unilateral entheseal ossification when it was significant as we described in the Methods section. The lack of a control group may be another limitation, especially for establishing differences with the ossification process related to age. Furthermore, BMI data are essential in this population since obesity has been directly linked to excessive bone formation, but we could not obtain this information from the medical records [32]. As for the strengths of the study, rigorous exclusion criteria were applied in order to distinguish the condition from spondyloarthropathies, which is often a considerable challenge. Furthermore, the proposal covers patients who are often underdiagnosed, and excluded from classifications and, consequently, from research studies.

## 6. Conclusions

Our findings suggest the presence of different phenotypes in DISH. We describe three patterns that differ in many aspects, not only age, gender or clinical symptoms, but also in radiographic signs. A new set of classification criteria need to be defined that consider the broad manifestations of hyperostotic disease in order to further our knowledge of the ossification process, to develop specific therapies, and to avoid disability or serious complications.

## Figures and Tables

**Figure 1 medicina-57-01005-f001:**
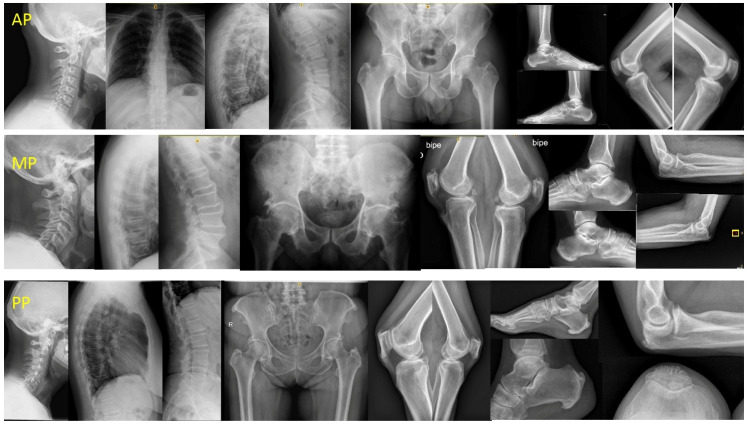
DISH patterns. Legend: AP: male 54 years, 10 years evolution. DISH signs in TL spine and only one enthesophyte (K) with no hip involvement. MP: male 70 years, 20 years evolution. Ankylosed column. Severe coxopathy. More than three peripheral enthesophytes (E, K, A). PP: female 68 years, 18 years evolution. No spine involvement, suspicion in CC. Almost all entheses affected (E, MT, K, A, C). DISH: Diffuse idiopathic skeletal hyperostosis; AP: axial pattern; MP: mixed pattern; PP: peripheral pattern; TL: thoracolumbar; E: elbows; MT: major trochanter; K: knees; A: Achilles; C: calcaneus; CC: cervical spine.

**Figure 2 medicina-57-01005-f002:**
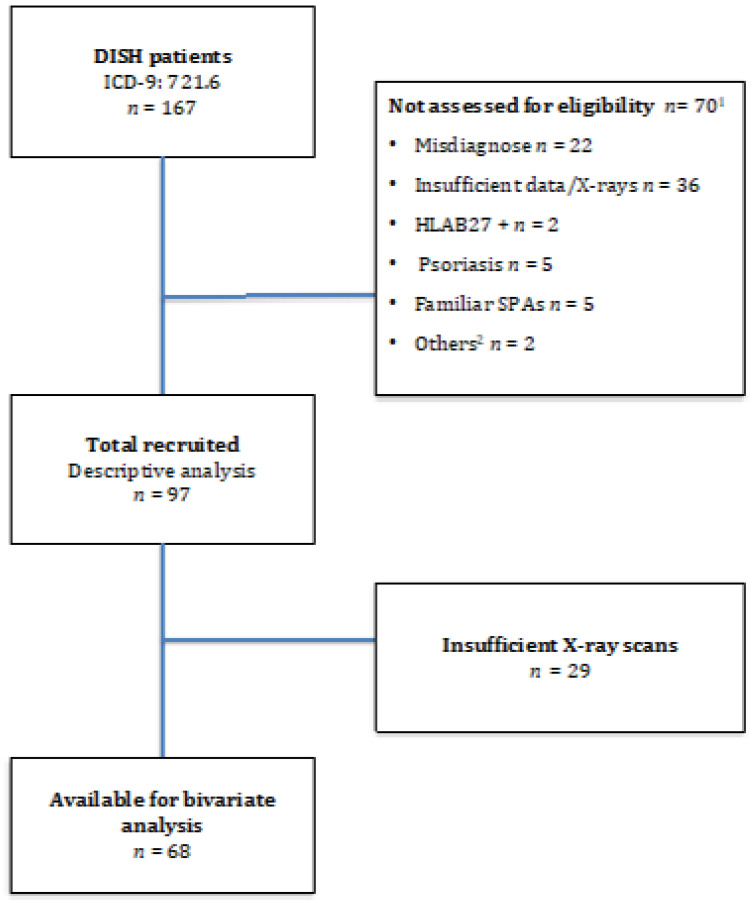
Flow Diagram. Legend: ^1^ 70 patients presented exclusion criteria and were not eligible for the study. ^2^ SPAS: Spondyloarthropathies. ^3^ Other reasons: undetermined uveitis, IBD background.

**Table 1 medicina-57-01005-t001:** Characteristics of the Study Sample (*n* = 97).

Variable	
Sex, *n* (%)	
Females	41 (42.3)
Males	56 (57.7)
Age at diagnosis, years	65.6 (9.5)
Age at onset, years	58.2 (9.0)
Comorbidities, *n* (%)	
Type 2 diabetes	
Yes	24 (25.0)
No	72 (75.0)
Dyslipidaemia	
Yes	35 (36.5)
No	61 (63.5)
CVE ^1^	
Yes	11 (11.5)
No	85 (88.5)
AHT ^2^	
Yes	49 (51.0)
No	47 (49.0)
Resnick criteria, *n* (%)	
Yes	70 (72.2)
No	27 (27.8)
Utsinger criteria, *n* (%)	
Definite	70 (72.2)
Probable	20 (20.6)
Possible	7 (7.2)
Presenting symptoms, *n* (%)	
Pain/limitation TS ^3^/LS ^4^	42 (43.8)
Enthesitis	23 (24.0)
Radiological findings	10 (10.4)
Hip pain/limitation	5 (5.2)
Pain/limitation CS ^5^	16 (16.7)
Enthesitis, *n* (%)	
Yes	39 (40.2)
No	58 (59.8)
Patterns, *n* 68 (%)	
Peripheral	20 (29.4)
Axial	21 (30.9)
Mixed	27 (39.7)
Axial involvement *n* (%)	
CS ^5^ (*n* = 64)	
Affected	37 (57.8)
Not affected	27 (42.2)
TS ^3^ (*n* = 92)	
Affected	85 (92.4)
Not affected	7 (7.6)
LS ^4^ (*n* = 89)	
Affected	50 (56.2)
Not affected	39 (43.8)
Extraspinal involvement, *n* (%)	
Elbow [olecranon] (*n* = 44)	
Affected	30 (68.2)
Not affected	14 (31.8)
Hip [acetabular excrescences] (*n* = 82)	
Affected	42 (51.2)
Not affected	40 (48.8)
Major Trochanter (*n* = 80)	
Affected	65 (81.3)
Not affected	15 (18.8)
Knees (sup/inf patella, tibial tuberosity) (*n* = 65)	
Affected	43 (66.2)
Not affected	22 (33.8)
Achilles tendon insertion (*n* = 54)	
Affected	43 (79.6)
Not affected	11 (20.4)
Plantar calcaneus (*n* = 52)	
Affected	39 (75.0)
Not affected	13 (25.0)

**Legend:** Values are presented with mean (SD). ^1^ CVE: cardiovascular events; ^2^ AHT: arterial hypertension; ^3^ TS: thoracic spine; ^4^ LS: lumbar spine; ^5^ CS: cervical spine.

**Table 2 medicina-57-01005-t002:** Comparison of variables between dish clinicoradiological patterns (*n* = 68).

PATTERNS TYPE	PP		AP		MP		*p*-Value
Age Dg ^1^, years (*n*/mean/SD)	20	62.05 (11.81)	21	68.14 (8.76)	27	66.30 (8.19)	0.119
Age at onset, years (*n*/mean/SD)	17	53.88 (5.31) ^#^	15	60.53 (10.27)	21	60.05 (7.36) ^#^	**0.036**
Sex (*n*/%)							**0.004**
Females	14	70.0% ^a^	4	19.0% ^b^	10	37.0% ^a,b^	
Males	6	30.0% ^a^	17	81.0% ^b^	17	63.0% ^a,b^	
Diagnostic delay, years (*n*/mean/SD)	17	5.53 (8.83)	15	7.33 (8.16)	21	6.52 (8.24)	0.832
Comorbidities (*n*/%)							
Type 2 diabetes							0.419
Yes	3	15.8%	7	33.3%	8	29.6%	
No	16	84.2%	14	66.7%	19	70.4%	
Dyslipidaemia							0.739
Yes	6	31.6%	9	42.9%	11	40.7%	
No	13	68.4%	12	57.1%	16	59.3%	
CVE ^2^							**0.001**
Yes	1	5.3% ^a,b^	7	33.3% ^b^	0	0.0% ^a^	
No	18	94.7% ^a,b^	14	66.7% ^b^	27	100.0% ^a^	
AHT ^3^							0.123
Yes	7	36.8%	8	38.1%	17	63.0%	
No	12	63.2%	13	61.9%	10	37.0%	
Presenting symptom (*n*/%)							**0.007**
Pain/limitation TS ^4^/LS ^5^	4	20.0% ^a^	10	47.6% ^a^	10	38.5% ^a^	
Clin enthesopathy	11	55.0% ^a^	1	4.8% ^b^	9	34.6% ^a^	
Rx findings ^6^	1	5.0% ^a^	6	28.6% ^a^	1	3.8% ^a^	
Hip pain/limitation	2	10.0% ^a^	1	4.8% ^a^	1	3.8% ^a^	
Pain/limitation CS ^7^	2	10.0% ^a^	3	14.3% ^a^	5	19.2% ^a^	
Clinical enthesopathy (*n*/%)							**<0.001**
Yes	17	85.0% ^a^	5	23.8% ^b^	11	40.7% ^b^	
No	3	15.0% ^a^	16	76.2% ^b^	16	59.3% ^b^	
Radiological findings (*n*/%)							
CS ^7^ (*n* = 45)							**0.020**
Affected	3	18.8% ^a^	6	60.0% ^a,b^	12	63.2% ^b^	
Not affected	13	81.3% ^a^	4	40.0% ^a,b^	7	36.8% ^b^	
TS ^4^ (*n* = 63)							0.059
Affected	15	78.9%	19	100.0%	24	96.0%	
Not affected	4	21.1%	0	0.0%	1	4.0%	
LS ^5^ (*n* = 63)							**<0.001**
Affected	3	15.0% ^a^	12	66.7% ^b^	19	76.0% ^b^	
Not affected	17	85.0% ^a^	6	33.3% ^b^	6	24.0% ^b^	
Hip (*n* = 63)							**0.035**
Affected	13	65.0% ^a^	5	26.3% ^b^	14	58.3% ^a,b^	
Not affected	7	35.0% ^a^	14	73.7% ^b^	10	41.7% ^a,b^	

**Legend:**^1^ Age at diagnosis; ^2^ CVE: cardiovascular events; ^3^ AHT: arterial hypertension; ^4^ TS: thoracic spine; ^5^ LS: lumbar spine; ^6^ radiological finding (asymptomatic patient); ^7^ CS: cervical spine; PP: peripheral pattern; AP: axial Pattern; MP: mixed Pattern. ^a,b^: Each subscript letter denotes a subset of pattern types categories whose column proportions do not differ significantly from each other at the 0.05 level. ^#^ multiple comparisons, Bonferroni method.

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
