# Peer review of "Identifying Clinicoradiological Phenotypes in Diffuse Idiopathic Skeletal Hyperostosis: A Cross-Sectional Study"

_medicina, 2021, doi:10.3390/medicina57101005_

Round 1

Reviewer 1 Report

In this observational study Clavaguera et. al. made important findings and provided insight into the different of phenotypes of Diffuse Idiopathic Skeletal Hyperostosis. I commend the authors for their interesting findings and ability to apply such rigorous comparisons between the three phenotypes.

I do have some specific points to address:

  1. It appears to be unclear. Did just the author TC make an assessment of the radiographs? It states that multiple individuals had insight in the diagnosis (line 98).
    1. It is customary to have two individuals independently assessing the radiographs and if any discrepancies occur, then a third person can assess.
    2. Given the nature of the topic and the author’s claim that there are 3 different phenotypes, I must insist another person assess the radiographs to ensure no patients are excluded.
  2. There are 22 patients who were misdiagnosed. I would be interesting to include what their primary diagnosis was. A simple line in the methods would add clarity to the paper and aid future clinicians to not misdiagnosis DISH.
  3. In assessing the radiographs, was there any differences noted between Axial and Peripheral hyperostosis? This could indicate that there are two differences between lesion composition and the potential mechanism of disposition.
  4. I find it interesting that there was no CVE in the mixed population vs the 33% in the axial population. I know its hypothesized that this could be due to sample size but its still interesting. Could any other hypothesis be put forward in the discussion.

Author Response

Response to Reviewer 1 Comments:

POINT 1: It appears to be unclear. Did just the author TC make an assessment of the radiographs? It states that multiple individuals had insight in the diagnosis (line 98).

Response 1: Yes, different physicians from rheumatology, orthopaedic or rehabilitation departments could establish the diagnosis but TC (me) assessed if patients met all the requirements (diagnostic, inclusion and exclusion criteria).

POINT 2: It is customary to have two individuals independently assessing the radiographs and if any discrepancies occur, then a third person can assess.

Response 2: Yes, I know it's a limitation of the study, but I couldn't get other experts on the subject.

POINT 3: Given the nature of the topic and the author’s claim that there are 3 different phenotypes, I must insist another person assess the radiographs to ensure no patients are excluded.

Response 3: I agree, I can include in the limitation of the study, but at that moment, I had no way to do it.

POINT 4: There are 22 patients who were misdiagnosed. I would be interesting to include what their primary diagnosis was. A simple line in the methods would add clarity to the paper and aid future clinicians to not misdiagnosis DISH.

Response 4: I agree. Many patients suffered from spondylosis with abnormalities in spine alignment.

POINT 5: In assessing the radiographs, was there any differences noted between Axial and Peripheral hyperostosis? This could indicate that there are two differences between lesion composition and the potential mechanism of disposition.

Response 5: I didn't find differences; many peripheral pattern patients had spinal abnormalities identical with the other phenotypes (mixt and axial). The only difference was the extensions of vertebral signs that in this type are absent or scarce. In my opinion, the elemental lesion is the same: the enthesis.

POINT 6: I find it interesting that there was no CVE in the mixed population vs the 33% in the axial population. I know it’s hypothesized that this could be due to sample size but it’s still interesting. Could any other hypothesis be put forward in the discussion?

Response 6: Again in my opinion, apart from statistical "accident", I think that CVE is more associated in the axial group because patients are older and there is a male gender predomination.

Reviewer 2 Report

nicely done paper - well written and interesting

Author Response

I think I have no request?

Reviewer 3 Report

I have no negative comments to the authors.

Author Response

I think there is no request..